# The Spatial Distribution of the House Mouse, *Mus musculus domesticus*, in Multi-Family Dwellings

**DOI:** 10.3390/ani12020197

**Published:** 2022-01-14

**Authors:** Shannon Sked, Chaofeng Liu, Salehe Abbar, Robert Corrigan, Richard Cooper, Changlu Wang

**Affiliations:** 1Department of Entomology, Rutgers-The State University of New Jersey, 96 Lipman Dr., New Brunswick, NJ 08901, USA; Shannon.Sked@rutgers.edu (S.S.); abbar.sally@gmail.com (S.A.); rcooper@sebs.rutgers.edu (R.C.); 2Department of Statistics, Purdue University, 250 N. University St, West Lafayette, IN 47907, USA; liu53@purdue.edu; 3RMC Pest Management Consulting, LLC., Briarcliff Manor, NY 10510, USA; cityrats@mac.com

**Keywords:** *Mus musculus domesticus*, spatial distribution, monitoring, apartment buildings

## Abstract

**Simple Summary:**

The management of house mice, *Mus musculus domesticus*, in low-income high-rise multi-family dwellings (MFDs) is often frustrated by the limited resources available through low-bid contracting. An improved understanding of the small-scale distribution of this important public health pest could allow the pest management industry to better allocate its limited time and resources to better managing infestations. This study utilized data from two research projects that measured house mouse infestation rates from four urban low-income MFDs to determine if a significant correlation between neighboring units exists in their infestation status. Results show that such a correlation exists whereby apartments that share a wall, ceiling or floor with a neighboring apartment that has a current infestation are more likely to have existing house mouse activity. This information can be utilized by the pest management industry to design monitoring strategies, during integrated pest management activities, to better ensure the elimination of house mice in low-income high-rise MFDs.

**Abstract:**

The house mouse, *Mus musculus domesticus*, creates significant public health risks for residents in low-income multi-family dwellings (MFDs). This study was designed to evaluate the spatial distribution of house mice in MFDs. Four low-income high-rise apartment buildings in three cities in New Jersey were selected for building-wide monitoring on two occasions with approximately one year between the monitoring events. The presence of a house mouse infestation was determined by placing mouse bait stations with three different non-toxic baits for a one-week period in all accessible units as well as common areas. Permutation tests were conducted to evaluate house mouse infestation spatial patterns. All four analyzed buildings exhibited a significant correlation between apartments with house mouse infestations and whether they share a common wall or ceiling/floor at both sampling periods except one building during the second inspection, which contained a high number of isolated apartments. Foraging ranges, speed of locomotion, and dispersal behavior of house mice are relatively larger, faster, and more common, respectively, compared to common urban arthropod pests. This could lead to the conclusion that house mice are as likely to infest non-neighboring apartments as those that share a wall or floor/ceiling. However, these results demonstrate that house mouse infestations tend to occur among apartments that share common walls or ceilings/floors. This spatial distribution pattern can be utilized in rodent management plans to improve the efficiency of house mouse management programs in MFDs.

## 1. Introduction

The cosmopolitan house mouse, *Mus musculus domesticus* (Schwarz and Schwarz, 1943) (Rodentia: Muridae), is a prevalent urban pest [1] in low-income communities with infestation rates among several studies in multi-family dwellings (MFDs) ranging from 36% to 49% [2,3,4]. This prevalence puts residents in these communities at risk due to house mice being of significant public health importance [5,6,7,8,9,10,11,12,13,14,15,16]. Their ability to live in close association with humans [1,17,18] increases that risk further. This commensal relationship among humans and house mice [19] along with the particular risk for those living in low-income communities calls for a better understanding of how house mice utilize space within a building so that the pest management industry can address infestations in a more economical manner for this subset of the population.

For house mice to successfully invade a new area, whether that area is an entire building or a previously uninfested apartment, population invasion and establishment must be supported by several factors [20]. Novel introductions require specific parameters for establishment and expansion to occur. Propagule pressure, in terms of both the number of individuals within a novel introduction and the number of introductions, must be enough for house mice to successfully invade and begin establishment in a new area. Establishment is unlikely without a significant number within a founder population [21,22], whether that number is created from a single introductory event or through an accumulation of multiple introductions. Often the causes of the unsuccessful establishment of founder populations are from Allee effects. Allee effects that prevent establishment include failure to locate mates [23], competitive feeding [24], genetic depression via inbreeding [25], and excessive depredation [26].

House mice have several population, behavioral, and physiological attributes that allow them to overcome these effects whether as a new introduction into a geographic area, into a new building, or from one apartment into a neighboring, previously uninfested apartment. Research has shown that house mice display behaviors that allow them to better avoid predators and expand their scouting range for increased food and mate finding opportunities. While house mouse activity, scent marking, and visitations did not appear to be affected on a large geographic scale when in the presence of a predator cue (cat urine), their activity was more clustered and dispersed at intermediate spatial scales as a result of the predator cue [27]. Laboratory studies found that newly introduced house mice will explore an entire area and feed from several locations made available with some preference observed on a few feeding locations [28]. When new trays were introduced, they exhibited neophilic feeding behaviors with familiar foods. In a field experiment, house mice that were introduced by sex in two separate areas on an island displayed a tenfold greater range expansion than what was found in the previously established population on that same island [29] Thereby, house mice exhibit behaviors that allow for rapid range expansion to investigate and find new sources of familiar foods as well as locate suitable mates for population growth to occur; both of which allow house mice to overcome specific Allee effects for successful establishment. This has also been demonstrated looking at historical data using phylogeographic and population genetics analyses [30]. Lippens et al. (2017) demonstrated that the invasion and establishment of house mice was very complex, with multiple founder effects established, and dispersal through human mediation well documented. These studies clearly show that house mice are capable of altering their behaviors when establishing a new population or expanding their territories into new areas.

What is not understood, is if house mice use these behaviors within a building to expand their range between neighboring units or from one distinct area of an MFD to another. What is known is that house mice are able to establish populations in close proximity to people worldwide [1,10,18] and exploit human activities to support their populations [17], as humans create resource opportunities for rodents to exploit [31]. Once established, they create a tight social network, or a “deme” [32]. Each deme is typically within a ~3–10 m [33] range with densities as high as 70/m [2,12]. The deme utilizes resource availabilities to determine territorial defense areas [34]. Eventually, a deme will expand their populations through territorial budding [35] driven by aggressive behaviors of dominant males towards younger, non-dominant males [12,36,37], which begin novelty-seeking exploratory behaviors and ultimately find new nest locations [38].

While this is an overarching understanding of how demes function in general, there is much complexity to this system as individuals’ behaviors and territorial scouting become dynamic as nests mature [39] and is driven greatly by resource availability [40]. With a fast post-partum oestrus period of 12–18 h, fast rates of sexual maturation, and a short, 19–21 d gestation period, the fecundity rate of house mice is high [33]. This affords a fast rate of population growth and persistent nest budding to occur, provided that resources remain available for the growing population via rapid expansion of young males [41]. Often, multiple infestations of apartments within one building develop from a related group of demes as genetic studies have demonstrated that house mouse populations are related at the building scale [13]. At larger geographic scales, house mouse populations increase as the spread of urbanization along with food and habitat resources increases [42,43].

Within multi-family high-rise buildings, house mouse activity is closely associated with the quality of exclusion in place along the outer perimeter building envelope. In work previously conducted, the level of sanitation and clutter within individual apartments was not a significant factor in determining whether or not house mouse activity was found within apartments. However, there was a spatial relationship between the floor on which an apartment was located and the risk of an apartment having house mouse activity. The lower three floors of high-rise apartment buildings were more commonly found to have house mouse activity than floors above the third [44]. Therefore, there is a need to understand the building-wide spatial distribution and movement of house mice in apartment buildings that have infestations. House mouse integrated pest management programs (IPM) should include accurate identification of all apartments with mouse activity within a building. Understanding house mouse spatial distribution and movement will afford the pest management industry the ability to target and treat infestations in a manner that is less labor intensive and can reduce the risks associated with treatment strategies to residents and staff. Additionally, public health risks and zoonotic disease risks associated with house mouse infestations can also be reduced. An improved understanding of where house mice are more likely to occur from an original infestation source could promote proactive management practices, further improving eradication efforts and reducing risks.

We subjected data from previous studies [44,45], where building-wide inspections were conducted, to binomial permutation tests to understand how house mice utilize space in low-income MFDs. The objective of this study was to evaluate the risks associated with infested neighboring units between apartments within a building. A better understanding of house mouse distribution among neighboring units should afford better monitoring practices for house mice. We anticipate this information will aid the pest management industry in being able to utilize precision management tactics, through targeting monitoring where infestation risks are likely high. This is especially important in low-income communities where budget constraints often prevent comprehensive house mouse management programs with limited time constraints for services.

## 2. Materials and Methods

### 2.1. Study Sites

This study was conducted at four low-income high-rise apartment buildings in three cities in New Jersey, USA. One building (T_1_) was located in Trenton, New Jersey; one building (L_1_) was located in Linden, New Jersey; while the remaining two buildings (P_1_ and P_2_) were located in Patterson, New Jersey. The number of apartments in T_1_, L_1_, P_1_, and P_2_ were 246, 200, 96, and 96, respectively. The proportion of one-bedroom, two-bedroom, and studio or efficiency apartments at T_1_ was 76%, 18%, and 6%, respectively. L_1_ consisted of 60% studio apartments and 40% one-bedroom apartments with no two-bedroom units. P_1_ and P_2_ contained 15, 56, and 29% of studio, one-bedroom, and two-bedroom apartments, respectively. The number of floors at T_1_, L_1_, P_1_, and P_2_, were 15, 11, 7, and 7, respectively.

All four buildings were constructed of mortar and brick on mortar outer perimeter walls, with interior walls constructed from dry wall on wood beams. This type of building construction is representative of high-rise low-income MFDs that began to dominate public housing in the 1950s [46]. While there were construction differences between buildings, we did not analyze the construction style of each individually as the four replicates were representative of common public housing multi-family buildings found in New Jersey. Each floor in all buildings had a trash chute that led to a central trash compactor room on the first floor. All buildings had common spaces including community areas and mechanical and boiler rooms; however, these spaces were not included in this study.

In buildings T_1_, L_1,_ and P_2_, the majority of apartments were adjoining with a shared wall or ceiling/floor to at least one other apartment. Building P_1_ had several apartments that were adjoined to a stairwell, elevator shaft, or open space, rather than another apartment, so that shared walls of those units were restricted to the ceiling-floor juncture. Table 1 shows the percent of apartments in each building that had two shared walls, one shared wall, or no shared walls with neighboring units. T_1_ and L_1_ had no apartments that were completely isolated with no shared walls with other apartments. However, P_1_ and P_2_ had a high portion of apartments that were isolated, 22% and 29%, respectively, sharing no walls with other apartments. The apartments with no shared walls with neighboring apartments, shared a wall with a stairwell, an elevator shaft, or a trash chute room. In total, there were 335 (52%) apartments with two shared walls with other apartments, 254 (40%) apartments with one shared wall with another apartment, and 49 (8%) apartments that were isolated.

### 2.2. Building-Wide Monitoring

Each of the four buildings were monitored two times with approximately 12 to 15 months between inspections. The study period occurred between July 2018 and March 2020. Protecta^®^ EVO^®^ Mouse bait stations (Bell Laboratories, Inc., Madison, WI, USA) with blank baits were installed in each accessible apartment during each inspection. Two stations were installed in each apartment at T_1_ and L_1_, with one station located near the stove and the second one along the baseboard adjacent to the heating system. One station was installed in each apartment at P_1_ and P_2_, located in the kitchen area near the stove. The two blank commercial baits included Detex^®^ Soft Bait with Lumitrack (Bell Laboratories, Inc., Madison, WI, USA) and Liphatech^®^ Rat & Mouse Attractant™ (Liphatech, Inc., Milwaukee, WI, USA). In addition to the commercially available non-toxic census baits, three ~1 g dollops of Hershey’s Spreads Chocolate (The Hershey’s Company, Hershey, PA, USA) was applied inside each bait stations as a novel food bait. Adding chocolate bait was based on a previous study that demonstrated a high percentage of mouse infestations would be missed by only using commercial rodent blank baits. One week after installing non-toxic bait stations, the stations were inspected for feeding activity by researchers and the results recorded. If any feeding was found on the baits, the respective apartment where the station was located was considered active for house mice.

House mouse management varied between inspections among the four buildings included in this study. Buildings at T_1_ and L_1_ were included in an IPM study during which Rutgers University researchers conducted IPM treatments in apartments with confirmed house mouse activity over the initial 12 weeks following the first building-wide inspection. After the 12-week IPM program was completed, house mouse management was conducted by a contracted pest control company at T_1_ and was conducted by licensed in-house staff at L_1_. At P_1_ and P_2_, house mouse management was conducted by a pest control contractor. Pest control contractors managed house mice in all four buildings primarily through responding to residential complaints or house management staff requests on an as-needed basis.

### 2.3. Data Analysis

Building layouts were drawn for the visual rendering of feeding activity and relative location and clustering. This was conducted for each building and each monitoring event. The data was then organized into a binomial data matrix to indicate the relationship between all apartments as pairs. Paired apartments with a shared wall or ceiling/floor were indicated as neighbors with a “1” and those without shared walls or a ceiling/floor were identified as independent with a “0”. This data was then analyzed based on the spatial structure matrix of the building and the infested connected apartment pairs. The independence of house mouse infestation and apartment connection was tested using a permutation test as follows: (1) Calculate the connected apartment pairs with infestation for each pair of connected apartments, if both are infested then count it as 1. Otherwise count it as 0. Then, sum the scores of all the pairs of connected apartments with infestation. (2) Permutate the infested apartments among all the apartments in the building and calculate the scores for each permutation of infestation in the building. (3) Calculate the upper quantiles of the observed score in Step 1 among the scores of permutations in Step 2 as the *p* value. A *p* value of <0.05 indicates significant correlation between neighboring units in their infestation status. The analysis was conducted using R statistical software (version 4.0.5, Veinna, Australia) [47].

## 3. Results

During the initial inspection at each building, researchers accessed 92%, 86%, 97%, and 94% of the apartments at T_1_, L_1_, P_1,_ and P_2_, respectively. During the second inspection, 82%, 89%, 94%, and 88% of the apartments were accessed at T_1_, L_1_, P_1,_ and P_2_, respectively. Visual representations of rodent feeding activity for the buildings on each inspection is shown in Figure 1a–h. House mouse infestations by building on each inspection occurrence, and the proportions of infestations in apartments that shared a wall or ceiling/floor, are summarized in Table 2. Building infestation rates, based on the apartments accessed during the first inspection, in T_1_, L_1_, P_1,_ and P_2_ were 8, 28, 18, and 19%, respectively. Infestation rates found during the second inspection in T_1_, L_1_, P_1,_ and P_2_ were 12, 2, 21, and 46%, respectively. Three of the four buildings had increased infestation rates between the two inspections. Only L_1_ showed a decrease in the proportion of apartments with feeding activity, from 28% during the initial visit to 2% during the second visit (Figure 1c,d).

When comparing whether or not apartments adjacent to infested apartments were more likely to have infestations, a correlation was demonstrated between neighboring units in their infestation status in T_1_, L_1,_ P_1,_ and P_2_ during the initial inspection (*p* < 0.001, *p* = 0.001, *p* = 0.001, and *p* = 0.004, respectively). This was also true in T_1_ and P_1_ during the second inspection (*p* < 0.001 and *p* = 0.03, respectively). Building L_1_ had only three infestations during the second monitoring event and therefore could not be tested.

During the second inspection, building P_2_ did not show a correlation between neighboring units in their infestation status (*p* = 0.09). This building had the lowest proportion of apartments with shared walls (Table 1) due to the construction infrastructure between several of the apartments on each floor such as stairwells, elevator shafts, and open spaces. In P_2_, 29% of the apartments were isolated without shared vertical walls (Figure 1g,h) as compared to 22% of the apartments in P_1_ being isolated (Figure 1e,f) and none of the apartments being isolated in T_1_ or L_1_ (Figure 1a–d).

The correlated distribution of the infested units in each building is further shown in Figure 2. L_1_ was not analyzed on the second (1-year) inspection since there was no infestation pairs with only three apartments infested.

## 4. Discussion

This study demonstrated that a correlation exists between the presence of house mouse infestation in a given apartment and the likelihood of house mouse infestations in neighboring units. If an apartment within an MFD building is confirmed to have house mouse activity, there is a significantly higher chance that the adjacent apartments with shared walls or shared ceilings/floors are also experiencing house mouse activity. Younger male mice are often the predominant individuals associated with initial dispersions [12,36,37,38] that can be characterized as being exploratory, following walls through thigmotaxis, visual landmarks, and odors produced by urine pheromones and plantar glands on the feet [48,49,50]. Ultimately, some mice establish new suitable nest locations. The high mobility of house mice facilitates exploratory scouts dispersing during these excursions [51,52]. This behavior and additional longer explorations, compared to most structural insect pests, facilitates the house mouse to utilize shared inter-apartment conduits and other points of egress in walls, floors, and ceilings, which ultimately expands their distribution between apartment units and throughout an MFD building easily. This is similar to that reported with other urban pests [53,54].

An alternate hypothesis for the correlation found between an apartment with a house mouse infestation and the likelihood of a neighboring unit experiencing an infestation, is a result of repeated invasions, rather than established deme expansion from an initial introduction. In this hypothesis, a novel introduction would establish a deme in one apartment and create territorial competition, causing subsequent introductions of non-associated house mice entering through the same entry points, to continue to move to the next suitable habitat such as a neighboring apartment. Both hypotheses would result in the findings presented in this study. It is also a logical possibility that a combination of both hypotheses (initial colonizing deme and repeated introductions) is occurring at the same time affording for the results observed. This study does not address whether either occurred independently since the number of introductions was not controlled, and therefore not tested. Additionally, the one-year time gap between sequential inspections is not frequent enough to look for novel introductions.

Resident complaints are not an accurate indicator of house mouse infestations in apartments in low-income MFDs [44]. Therefore, a house mouse management approach that utilizes reactive treatment approaches based on resident or staff complaints is likely to be inadequate in identifying and eliminating MFD house mouse infestations. Our research proposes that a proactive building-wide monitoring system identifies the specific locations of MFD house mouse infestations. By understanding the correlation risks associated with neighboring apartments, the full distribution of house mouse infestations in MFDs can be further understood and ultimately better managed. Monitoring should include a systematic approach that assesses neighboring units whenever house mice are confirmed to be active in any one apartment. Apartments on both sides, below, and above the infested unit should be monitored as well while pest management efforts are enacted. Doing so is critical in determining the breadth of the mouse activity.

Additionally, understanding the specific building construction and floorplan layout of an MFD is essential in evaluating the risks associated with the infested and proximal apartment dynamics as highlighted in this study, including non-apartment areas that could be adjacent to apartments such as trash chutes and stairwells. For example, in P_1_, it is probable that the high proportion of apartments that lacked shared neighboring walls, floors or ceilings, explain why they also lacked any significant common infestation relationship. This could also be true for apartments that are adjacent to trash chutes which offer food resources and movement pathways, although this was not evaluated in this study. Therefore, it would be important to monitor the structures adjacent to apartments that do not share walls with other apartments, such as stairwells, trash chutes and elevator shafts. The more isolated each apartment is in the layout, the lower the potential risk associated with neighboring apartments that have existing house mouse infestations. Yet, it is logical to assume that mice will use spaces adjacent to infested apartments as corridors to expand their distribution.

Finally, this research has important implications for the pest management industry. First, building-wide monitoring not only facilitates a better understanding of where house mouse infestations occur within an MFD during any one inspection event [44], but such monitoring also pinpoints those apartments subject to an increased risk from likely soon-to-be infestations. Second, expanding monitoring efforts to the proximal units and areas beyond an individual apartment, assists in identifying the critically important building-wide dispersal pathways. Third, monitoring proximal areas adjacent to apartments with known infestations can reveal existing infestations in unknown or hard-to-access locations and common infrastructural elements of MFDs such as stairwells, trash chutes, and elevator shafts.

## 5. Conclusions

While low-bid pest management contracts are typical for low-income communities, they usually do not afford the time to conduct full building inspections with every visit. However, by utilizing the outcomes produced in this study, pest management professionals can greatly increase their chances of locating and curtailing any spreading distributions of mice in MFDs where mice have already been found to exist. This information enables pest management professionals to balance the time necessary for the essential science-based inspections with the need to broaden inspection events beyond a reactive complaint-based program, focusing on individual apartments alone. By understanding where higher risks exist, proactive monitoring during follow up inspections reduces the labor requirement while fully eliminating infestations throughout the building. This in turn, significantly reduces the expensive and frustrating return visits (callbacks) that become necessary when rodent IPM tactics fail.

Finally, targeted treatments can be conducted where house mouse risks are the highest; making for an efficient reduction in the amount of rodenticides needed and thereby reducing the risks associated with unnecessary pesticide exposure. Additionally, targeted rodent IPM tactics are more likely to be successful in eliminating building-wide house mouse infestations, thereby reducing the risk of zoonotic diseases and public health risks associated with this important public health pest. Using the research results of this study, house mouse management can more successfully adhere to the principles of IPM and thus produce the highly desirable outcome of mouse management sustainability.

## Figures and Tables

**Figure 1 animals-12-00197-f001:**
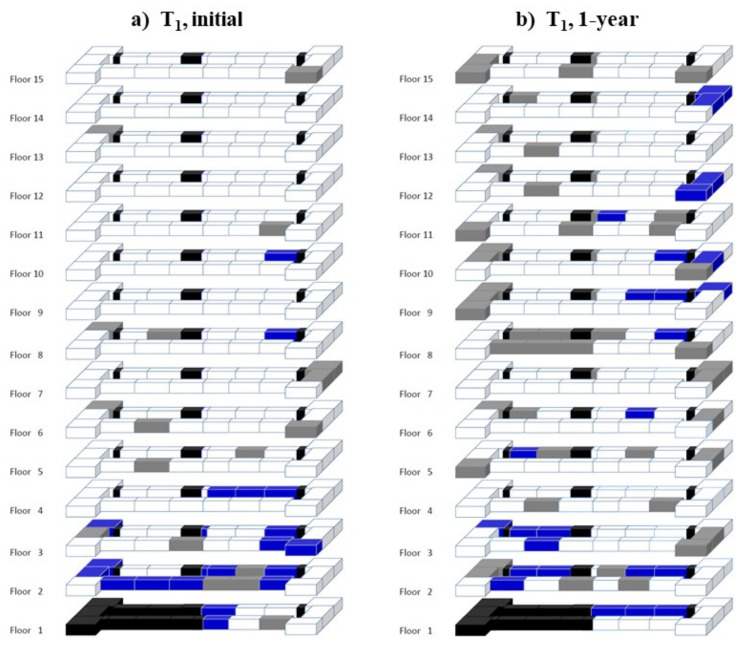
Representative map (not to scale) of the buildings monitored for house mouse activity. T_1_, L_1_, P_1_ and P_2_ represents buildings at Trenton, Linden, and two buildings at Paterson, respectively. Blue blocks indicate apartments with mouse activity. Black blocks indicate elevator, stairwell, or open space. Gray blocks indicate apartments where access was denied.

**Figure 2 animals-12-00197-f002:**
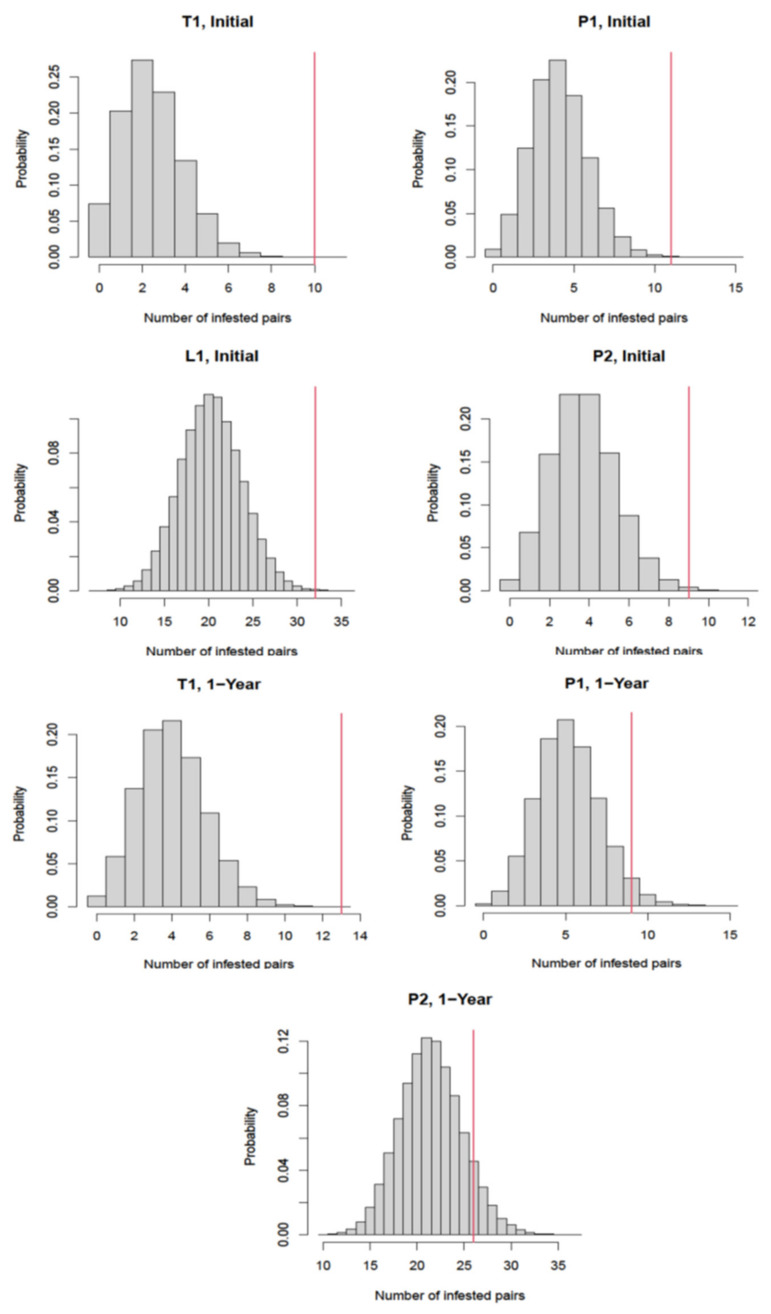
Distribution of the number of connected apartment pairs with both mouse infestation assuming no infestation correlation between two connected apartments in each building. The red vertical line indicates the actual number of infested connected apartment pairs in the building.

**Table 1 animals-12-00197-t001:** Percent of apartments in each building that had two shared walls, one shared wall, or no shared walls with other apartments in the building.

Building	Number of Apartments	Number of Floors	Percent Apartments with Two Shared Walls	Percent Apartments with One Shared Wall	Percent Apartments with No Shared Walls
T_1_	246	15	65%	35%	0%
L_1_	200	11	60%	40%	0%
P_1_	96	7	32%	46%	22%
P_2_	96	7	25%	46%	29%

**Table 2 animals-12-00197-t002:** Summary of apartments with house mouse infestations and the proportion of apartments with a shared wall or ceiling/floor.

Building	Inspection Occurrence	Number of Apartments Accessed and Inspected	Number of Apartments Infested	Infestation Rate of Apartments Inspected	Number (%) of Infested Apartments with Shared Walls or Ceiling/Floors
T_1_	Initial	226	19	8%	15 (79%)
	1-year	202	25	12%	18 (72%)
L_1_	Initial	172	49	28%	39 (80%)
	1-year	178	3	2%	0 (0%)
P_1_	Initial	93	17	18%	9 (53%)
	1-year	90	19	21%	11(58%)
P_2_	Initial	90	17	19%	14 (82%)
	1-year	84	39	46%	28 (72%)

## Data Availability

Data was collected and is held at the institution performing the study and analysis. Data is available for review upon request.

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
