# Peer review of "The Spatial Distribution of the House Mouse, Mus musculus domesticus, in Multi-Family Dwellings"

_animals, 2022, doi:10.3390/ani12020197_

Round 1
Reviewer 1 Report
This manuscript presents an interesting and novel study of house mouse movement and infestation dynamics. It provides valuable information for those involved in pest management or interested in the natural history of one of the worlds most common synanthropes. The study design, statistical analysis, and interpretation are sound. But, I think the manuscript can be improved in two major ways 1) an expansion of the analysis to better capture the spatial extent of the observed pattern, described below; 2) consideration of alternative hypotheses in the discussion, particularly given the authors own description of house mouse invasion dynamics described in the introduction, also described below. I also list several minor comments and suggestions below that should be addressed.
L62. Do the authors mean “allee effects”, rather than “allele effects”. Alleles refer specifically to genetic variants, but the description includes other factors better describes as “allee effects”
L74. “Engage” is an odd word choice here. Maybe “explore” or “move within” is more descriptive?
L92: Make “17” a bracketed citation
L108: remove apostrophe from “populations’”
L115: should be “the floor on which an apartment was located” or similar
L122-123: I would encourage an expansion of this idea, as it defines the relevance and importance of the study. I expect that understanding the spatial distribution and thereby better targeting infestations could decrease the likelihood of reinfestation, thus improving the effectiveness of the treatment and reducing overall costs and burden to residents. It may also allow relative risk of zoonoses to be better predicted spatially and promote preventative treatments rather than reactive treatments, which is a key component of IPM. Some of these ideas are presented in the last intro paragraph. I suggest keeping these thoughts in one paragraph together to improve impact.
L151-162: Is there information about the building material or construction style of each building. It may be relevant to note if materials differ in their potential permissiveness to mouse activity (e.g., brick vs. sheetrock). Were interstitial spaces noted or were walls solid? Some of this information is presented on lines 234-235.
Data Analysis: I think it would be interesting and relevant to extend this analytical process to apartments that are separated by more than wall. If we see a relationship among directly adjacent apartments, how far out does that relationship extend? Lines 275-278 give specific advice about neighboring apartments, but it would be useful to know whether other nearby apartments should be of similar or less concern. Lines 291-292 touch on the potential for movement to non-neighboring but nearby apartments. Why not analyze this? You could repeat the same statistical process but code those apartments separated by a single other apartment or infrastructure space or within, say 20 meters in any direction (define the distance based on the maps of the buildings). It would be useful in a management context to know if the higher risk exists only for the neighboring apartment or whether non-adjacent but nearby apartments also experience higher risk. I would continue this process iteratively, increasing the distance, until the relationship is no longer significant. I can envision a plot with distance on the x axis and relative increase in effect on the y-axis, which would likely decay with distance. This would be akin to a spatial autocorrelation analysis and/or correlogram often analyzed with the vegan or ecodist packages in R.
Figure1: Can we group these different schematics together on a single multipane figure. I would also like to see scale bars for each building (or one scale bar with schematics scaled to one another), which would improve interpretation.
L254: I would word this as: “This study demonstrated that correlations exist between the presence of house mouse infestations in a given apartment and the likelihood of house mouse infestation in a neighboring unit.” The term “house mouse infestation correlations” doesn’t really make sense, sounding more like a type of correlation.
Discussion: Given the introduction statements about initial building infestations often requiring sufficient initial colonization numbers, would it also be a reasonable hypothesis that neighboring infestations are possibly due to their proximity to points of initial entry/colonization rather than as a product of subsequent dispersal? Perhaps there is clustering of initial infestation locations that might cause the observed pattern. I would think this hypothesis would mostly be true for lower floor apartments or those near vertical movement corridors at points of entry, so perhaps the presence of this pattern at higher floors is good evidence against. I suspect there are not sufficient data to analyze these patterns in a by-floor manner, but it could be interesting to know if the pattern changes with elevation within MFDs.
One additional hypothesis would be repeated invasion by new groups of house mice, where initial colonizers would occupy the first suitable habitat they encounter, while territoriality and competition would cause subsequent colonizers to move to the next available habitat, presumably the neighboring apartment. This would be a form of ecological priority effect.
The authors may want to discuss these alternative hypothesis or other potential causes of clustering.
L315: I would put “callbacks” in quotations since it is an industry-specific term, or define somewhere within the manuscript. It is not common knowledge among this journal’s audience that mice in MFDs are closely associated with callbacks.
L317-321: I would also mention reducing the risk of zoonoses or public health impacts.
Author Response
Reviewers Comments for Manuscript Submission to Animals
The spatial distribution of the house mouse, Mus musculus domesticus, in multi-family dwellings
Shannon Leif Sked , Chaofeng Liu , Salehe Abbar , Robert Corrigan , Richard Cooper , Changlu Wang *
Reviewer #1 Comments:
This manuscript presents an interesting and novel study of house mouse movement and infestation dynamics. It provides valuable information for those involved in pest management or interested in the natural history of one of the worlds most common synanthropes. The study design, statistical analysis, and interpretation are sound. But, I think the manuscript can be improved in two major ways 1) an expansion of the analysis to better capture the spatial extent of the observed pattern, described below; 2) consideration of alternative hypotheses in the discussion, particularly given the authors own description of house mouse invasion dynamics described in the introduction, also described below. I also list several minor comments and suggestions below that should be addressed.
L62. Do the authors mean “allee effects”, rather than “allele effects”. Alleles refer specifically to genetic variants, but the description includes other factors better describes as “allee effects” I will correct this – as it should be allee effects. I thought I made this correction after Dr. Cooper advised earlier.
L74. “Engage” is an odd word choice here. Maybe “explore” or “move within” is more descriptive? Corrected to “explore” as a more appropriate word.
L92: (Now L93) Make “17” a bracketed citation Corrected
L108: remove apostrophe from “populations’” Corrected
L115: (Now L116) should be “the floor on which an apartment was located” or similar Corrected
L122-123: I would encourage an expansion of this idea, as it defines the relevance and importance of the study. I expect that understanding the spatial distribution and thereby better targeting infestations could decrease the likelihood of reinfestation, thus improving the effectiveness of the treatment and reducing overall costs and burden to residents. It may also allow relative risk of zoonoses to be better predicted spatially and promote preventative treatments rather than reactive treatments, which is a key component of IPM. Some of these ideas are presented in the last intro paragraph. I suggest keeping these thoughts in one paragraph together to improve impact. Added (L125-129) to explain the added benefits of proactive treatments and reducing zoonotic and public health risks.
L151-162: (Now L153-161) Is there information about the building material or construction style of each building. It may be relevant to note if materials differ in their potential permissiveness to mouse activity (e.g., brick vs. sheetrock). Were interstitial spaces noted or were walls solid? Some of this information is presented on lines 234-235. Added (L154-161) to describe the building construction; all representative of public housing multifamily buildings commonly found in New Jersey and therefore we didn’t evaluate differences among building construction in this study.
Data Analysis: I think it would be interesting and relevant to extend this analytical process to apartments that are separated by more than wall. If we see a relationship among directly adjacent apartments, how far out does that relationship extend? Lines 275-278 (Now L316-320) give specific advice about neighboring apartments, but it would be useful to know whether other nearby apartments should be of similar or less concern. Lines 291-292 (Now L333-336) touch on the potential for movement to non-neighboring but nearby apartments. Why not analyze this? You could repeat the same statistical process but code those apartments separated by a single other apartment or infrastructure space or within, say 20 meters in any direction (define the distance based on the maps of the buildings). It would be useful in a management context to know if the higher risk exists only for the neighboring apartment or whether non-adjacent but nearby apartments also experience higher risk. I would continue this process iteratively, increasing the distance, until the relationship is no longer significant. I can envision a plot with distance on the x axis and relative increase in effect on the y-axis, which would likely decay with distance. This would be akin to a spatial autocorrelation analysis and/or correlogram often analyzed with the vegan or ecodist packages in R. Added an analysis of correlation distribution by building and by inspection in Figure 2 with descriptive in Table 2 added.
Figure1: Can we group these different schematics together on a single multipane figure. I would also like to see scale bars for each building (or one scale bar with schematics scaled to one another), which would improve interpretation. Unfortunately, the size differences between buildings did not allow us to scale each to the same. These figures are representative of the building shapes and apartment sizes relative to one another in a building rather than a series of scale set CAD drawings. This is addressed in as making note that they are “not to scale” in the Figure legend. The figure has been reset for increasing the ease of view while following the results.
L254: (Now L273-275) I would word this as: “This study demonstrated that correlations exist between the presence of house mouse infestations in a given apartment and the likelihood of house mouse infestation in a neighboring unit.” The term “house mouse infestation correlations” doesn’t really make sense, sounding more like a type of correlation. Corrected sentence as recommended.
Discussion: Given the introduction statements about initial building infestations often requiring sufficient initial colonization numbers, would it also be a reasonable hypothesis that neighboring infestations are possibly due to their proximity to points of initial entry/colonization rather than as a product of subsequent dispersal? Perhaps there is clustering of initial infestation locations that might cause the observed pattern. I would think this hypothesis would mostly be true for lower floor apartments or those near vertical movement corridors at points of entry, so perhaps the presence of this pattern at higher floors is good evidence against. I suspect there are not sufficient data to analyze these patterns in a by-floor manner, but it could be interesting to know if the pattern changes with elevation within MFDs. While this is an interesting hypothesis, analyzing single floor data would likely not be a robust dataset to conclusively determine if this is occurring. We evaluated lower vs. upper floors in a prior study and did find that there was an effect where lower floors were more likely to have infestations. There is likely a combined effect of scouting behaviors leading to established dispersal patterns; however, this data does not address this specifically since the two variables were not able to be controlled to test for either effect independently.
One additional hypothesis would be repeated invasion by new groups of house mice, where initial colonizers would occupy the first suitable habitat they encounter, while territoriality and competition would cause subsequent colonizers to move to the next available habitat, presumably the neighboring apartment. This would be a form of ecological priority effect.
The authors may want to discuss these alternative hypothesis or other potential causes of clustering. Added a paragraph (L288-300) to describe both hypothesis for the cause of the results found and the limitations of the study in testing for either, independently.
L315: (Now L348) I would put “callbacks” in quotations since it is an industry-specific term, or define somewhere within the manuscript. It is not common knowledge among this journal’s audience that mice in MFDs are closely associated with callbacks. This was corrected with adding an explanation into the sentence (L347-349).
L317-321: (Now L350-352) I would also mention reducing the risk of zoonoses or public health impacts. We added a sentence (L353-355) to explain the inherent reduced risks with more effective rodent IPM tactics via targeted treatments.
Reviewer 2 Report
The study is focused on important and interesting topic of mice infestation in low-income multifamily dwellings.
The dataset obtained is quite large and have a great potential. By using permutation test was found, that flats adjacent to infested ones are more likely to be inhabited by mice. From different patterns of buildings are then in Discussion deduced why this finding is not true for all buildings. However, it can be analysed more ingeniously and included into statistics, as it is enabled by the dataset, which is not fully utilized. The infestation and position of each flat is known and could be detailed analysed. Results about infestation rate of different types of flats according to their position (isolated, one or two shared walls) or influence of floor should be added. It could be also analysed, if is more probable infestation through sharing flour or ceiling (= if infestation rate is more horizontally or vertically) or if flats near trash chutes are more infested.
In Material information in lines 141-147 should be included into Table.
More information (primarily about infestation rates of buildings and types of flats) from text of Results should be included into table, where it is better demonstrative and clear. In recent manuscript, data about infestation of particularly types of flats in each building are lacking.
More detailed information about types of flats and mice enemies should be added or discussed: if walls between flats are made from concrete or material which can be penetrated by mouse, if rat occurrence is supposed (it can influence mouse infestation), influence of house cats, etc.
Do you think (and it could be discussed), that mouse infestation in second session is due to new colonization from outside or due to surviving mice which remained in some flats/shared areas- it can be derived from position and number of floor of infested flats.
Author Response
Reviewers Comments for Manuscript Submission to Animals
The spatial distribution of the house mouse, Mus musculus domesticus, in multi-family dwellings
Shannon Leif Sked , Chaofeng Liu , Salehe Abbar , Robert Corrigan , Richard Cooper , Changlu Wang *
Reviewer #2 Comments:
The study is focused on important and interesting topic of mice infestation in low-income multifamily dwellings.
The dataset obtained is quite large and have a great potential. By using permutation test was found, that flats adjacent to infested ones are more likely to be inhabited by mice. From different patterns of buildings are then in Discussion deduced why this finding is not true for all buildings. However, it can be analysed more ingeniously and included into statistics, as it is enabled by the dataset, which is not fully utilized. The infestation and position of each flat is known and could be detailed analysed. Results about infestation rate of different types of flats according to their position (isolated, one or two shared walls) or influence of floor should be added. It could be also analysed, if is more probable infestation through sharing flour or ceiling (= if infestation rate is more horizontally or vertically) or if flats near trash chutes are more infested. Added an analysis of correlation distribution by building and by inspection in Figure 2 with descriptive in Table 2 added.
In Material information in lines 141-147 should be included into Table. Added the number of apartments and number of floors to Table 1.
More information (primarily about infestation rates of buildings and types of flats) from text of Results should be included into table, where it is better demonstrative and clear. In recent manuscript, data about infestation of particularly types of flats in each building are lacking. We did not capture data about infestations between different flat types; rather each was given a binomial rating of infested or non-infested. Since this study evaluated only movement, rather than impacts of apartment features, we did not record this information here.
More detailed information about types of flats and mice enemies should be added or discussed: if walls between flats are made from concrete or material which can be penetrated by mouse, if rat occurrence is supposed (it can influence mouse infestation), influence of house cats, etc. Added (L154-161) to describe the building construction; all representative of public housing multifamily buildings commonly found in New Jersey and therefore we didn’t evaluate differences among building construction in this study.
Do you think (and it could be discussed), that mouse infestation in second session is due to new colonization from outside or due to surviving mice which remained in some flats/shared areas- it can be derived from position and number of floor of infested flats. Added discussion on alternative hypothesis as to the cause of the correlation found and assumptions on each potential cause (L288-300)

Reviewer 3 Report
This study is undoubtedly of great practical importance and is of obvious interest for city services engaged in deratization. Although the scientific significance is not so high, this work may be published in the journal "Animals" because of its great practical importance.
The materials and methods indicate that the results of the work were subjected to statistical processing. However, the results of statistical processing are presented in a fragmented way. Though the projections of houses and their infestation of mice on Fig. 1 look quite interesting, it is necessary to have a table or tables in which the reliability of all data on the basis of which conclusions are made will be presented.
Author Response
Please find attached the responses to comments. Thank you for your time reviewing this manuscript.

Reviewer 4 Report
Dear Authors,
I believe your manuscript: 'The spatial distribution of the house mouse, Mus musculus domesticus, in multi-family dwellings' is an interesting work for the readers of 'Animals'. It is a well-structured, rather original work that obtains useful results for the management and control of Mus musculus domesticus in an urban environment. The statistical processing of the results is very good.
One thing I would like to say is the advice to improve Fig. 1 in some way. Although sufficiently clear, it is difficult to read, perhaps also due to its division into parts. It would be better to find another solution to illustrate the same results in another graphic format.
All the best
Author Response

(The authors gave the same response as above.)

Round 2
Reviewer 2 Report
I highly appreciated the adding of the last column to the Table 2. The Figure 2 seems a bit superfluous and complicated.
For better visibility and clearance, I propound to add legends (the name of building and trapping session) to Figure 1 directly to the each figure (however I do not know, if it is allowed due to technical or editorial rules).
There is a typo in Line 43- the species should be described by name of discoverer and year, this should be in branches- „ Mus musculus domesticus (Schwarz and Schwarz, 1943) “.
I have no other comments to the manuscript.
Author Response
Reviewers Comments for Manuscript Submission to Animals
The spatial distribution of the house mouse, Mus musculus domesticus, in multi-family dwellings
Shannon Leif Sked , Chaofeng Liu , Salehe Abbar , Robert Corrigan , Richard Cooper , Changlu Wang *
Reviewer #2 Second Set Comments:
I highly appreciated the adding of the last column to the Table 2. The Figure 2 seems a bit superfluous and complicated. We are glad that Table 2 has added value to the information presented. Figure 2 was added as we were requested by another reviewer to better describe the data. Figure 2 was the simplest way to present how the data were analyzed to satisfy that request.
For better visibility and clearance, I propound to add legends (the name of building and trapping session) to Figure 1 directly to the each figure (however I do not know, if it is allowed due to technical or editorial rules). We have added the building name and inspection period time to each building image in Figs. 1a-h.
There is a typo in Line 43- the species should be described by name of discoverer and year, this should be in branches- „ Mus musculus domesticus (Schwarz and Schwarz, 1943) “. This is corrected.
I have no other comments to the manuscript.